# Is Texture Predictive for Age and Sex in Brain MRI?

**Nick Pawlowski**[1]                                                     N.PAWLOWSKI16@IMPERIAL.AC.UK

**Ben Glocker**[1]                                                             B.GLOCKER@IMPERIAL.AC.UK

[1] *Biomedical Image Analysis Group, Imperial College London, UK*

## Abstract

Deep learning builds the foundation for many medical image analysis tasks where neural networks are often designed to have a large receptive field to incorporate long spatial dependencies. Recent work has shown that large receptive fields are not always necessary for computer vision tasks on natural images. We explore whether this translates to certain medical imaging tasks such as age and sex prediction from a T1-weighted brain MRI scans.

**Keywords:** Age and Sex Prediction, Brain MRI, Neural Networks

## 1. Introduction

Deep learning has become the de-facto standard method in many computer vision and medical image analysis applications (Litjens et al., 2017). Recently introduced BagNets (Brendel and Bethge, 2019) have shown that on natural images, neural networks can perform complex classification tasks by only interpreting texture information rather than global structure. BagNets interpret a neural network as a bag-of-features classifier that is composed of a localised feature extractor and a classifier that acts on the average bag-encoding.

We explore whether texture information is sufficient for certain tasks in medical image analysis. For this, we generalise BagNets to arbitrary regression tasks and 3D images and examine the performance of different receptive fields. We apply BagNets to age regression and sex classification tasks on T1-weigthed brain MRI to examine the dependency of modern deep learning architectures on local texture in these medical imaging tasks. We find that the bag-of-local-features approach yields comparable results to larger receptive fields.

## 2. Method

BagNets (Brendel and Bethge, 2019) are adaptations of the ResNet-50 architecture (He et al., 2016), that restrict the receptive field by replacing $3 \times 3$ convolutional kernels with $1 \times 1$ kernels. A regular ResNet-50 has a receptive field of 177 pixels, whereas BagNets explore receptive fields of 9, 17 and 33 pixels. The use of small receptive fields enforces locality in the extracted features. After extracting local features a global spatial average builds the bag-of-local-features and enforces the invariance to spatial relations. The bag of features is then processed by a linear layer to provide the final prediction. Because of the linearity of the average operation and the final linear layer, it is possible to exchange the order of those operations, which enables the extraction of localised prediction maps.

## 3. Experiments & Results

We test the BagNets on the public Cambridge Centre for Ageing and Neuroscience (Cam-CAN) dataset (Taylor et al., 2017). The dataset contains T1- and T2-weighted brain MRI of 652 healthy subjects within an age range of 18 to 87. We only use the T1-weighted scans for our experiments and randomly split the scans into training, validation and test sets with 456, 65 and 131 subjects each. All scans have an isotropic resolution of 1 mm. We use skull-stripped, bias-fiel corrected images and extract random crops of shape [128×160×160] during training. We whiten the images with statistics extracted from within the brain mask.

We use the architecture from (Brendel and Bethge, 2019) but replace 2D with 3D convolutions and half the number of feature maps. We train the network with batch size 1 and accumulate gradients over 16 batches. To alleviate the effects of small batches we use instance normalization (Ulyanov et al., 2016) instead of batch normalization (Ioffe and Szegedy, 2015). We use a cross-entropy loss for the sex classification and MSE loss for the age regression. We use the Adam optimizer (Kingma and Ba, 2014) with a learning rate of $\eta = 0.001$, $\epsilon = 10^{-5}$ and employ an $L_2$-regularization of $\lambda = 0.0001$. We train the network for 500 epochs and decay the learning rate by a factor of 10 every 100 epochs. We use the checkpoint with the best validation performance for evaluation on the test data.

Table 1 shows the mean absolute error (MAE) and accuracy for the age and sex prediction for different receptive fields. We achieve a MAE between $3.86 - 5.53$ years for age and an accuracy between $80.9 - 84.0\%$ for sex. Age regression has a stronger dependency on the receptive field than the sex classification. However, we find that the larger receptive field exhibits better training performance and might be prone to overfitting.

Table 1: Results for the age regression and sex classification task for different receptive fields. We report the mean absolute error for age and classification accuracy for sex. We find that small receptive fields yield comparable results on those tasks.

| Receptive Field | Age | Sex |
|---:|:---:|:---:|
| $(9mm)^3$ | 5.53 | 83.2% |
| $(17mm)^3$ | 5.32 | 84.0% |
| $(33mm)^3$ | 4.98 | 84.0% |
| $(177mm)^3$ | 3.86 | 80.9% |

We examine the local predictions on two examples from the test set in Figure 1 for age regression and Figure 2 for sex classification. The sex classification predicts 0 for male and 1 for female. The first row shows a 20 year old male subject, the second row shows an 80 year old female. The columns respectively show the middle slice of the T1-weighted MRI, the local predictions with receptive fields 9, 17, 33 and 177. Similarly to (Brendel and Bethge, 2019), we find that small receptive fields lead to more localised predictions, whereas larger receptive fields show more spread out predictions. Interestingly, the age regression exhibits very high variance predictions, where only few very high values contribute to the mean prediction of the volume. Generally, we find that the local predictions we get from our model do not seem as interpretable as in (Brendel and Bethge, 2019).

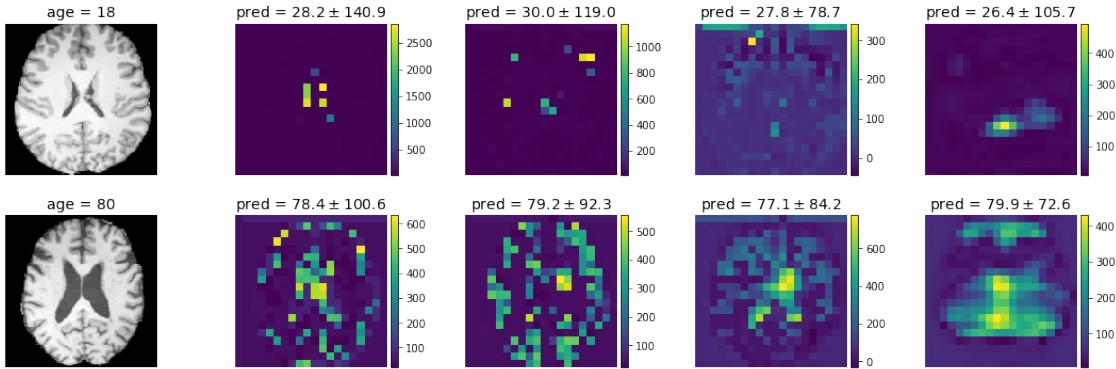

Figure 1: Localised age prediction on a 20 year old male subject (first row) and 80 year old female subject (second row). The columns show the middle slice of the T1-weighted MRI and the localised predictions for receptive fields 9, 17, 33 and 177.

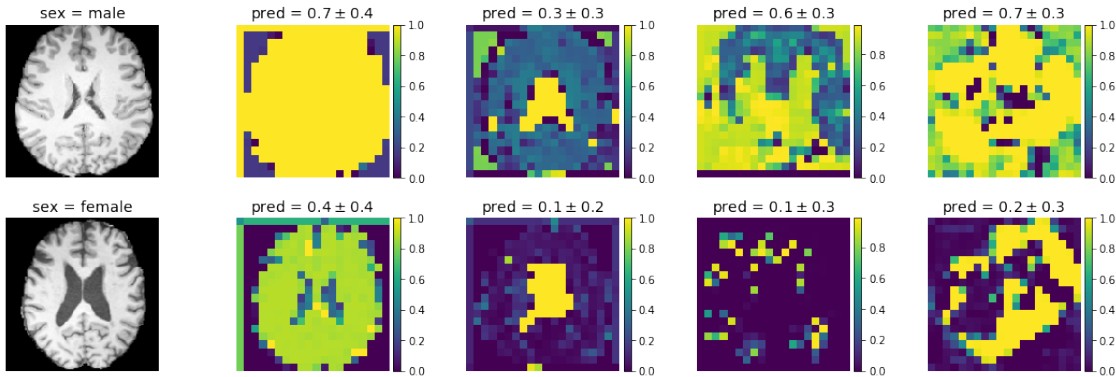

Figure 2: Localised sex classification on a 20 year old male subject (first row) and 80 year old female subject (second row). The different columns show the middle slice of the T1-weighted MRI and the localised predictions for receptive fields 9, 17, 33 and 177. The network predicts male as 0 and female as 1.

## 4. Discussion & Conclusion

We have generalised the concept of BagNets (Brendel and Bethge, 2019) to the setting of 3D images and general regression tasks. We have shown that a BagNet with a receptive field of $(9mm)^3$ yields surprisingly accurate predictions of age and sex from T1-weight MRI scans. However, we find that localised predictions of age and sex do not yield easily interpretable insights into the workings of the neural network which will be subject of future work. Further, we believe that more accurate localised predictions could lead to advanced clinical insights similar to (Becker et al., 2018; Cole et al., 2018).

## Acknowledgments

NP is supported by Microsoft Research PhD Scholarship and the EPSRC Centre for Doctoral Training in High Performance Embedded and Distributed Systems (HiPEDS, Grant Reference EP/L016796/1). BG received funding from the European Research Council (ERC) under the European Union's Horizon 2020 research and innovation programme (grant agreement No 757173, project MIRA, ERC-2017-STG). We gratefully acknowledge the support of NVIDIA with the donation of one Titan X GPU.

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
