# OpenReview forum: "Is Texture Predictive for Age and Sex in Brain MRI?"
_MIDL.io/2019/Conference/Abstract — MIDL Abstract 2019_

### Official Review · AnonReviewer2 · 2019-04-28
**improve comparative study**

**Rating:** 3
**Confidence:** 3

**Review:**

Is texture predictive detection and grading of prostate cancer in multi-parametric MRI?

This paper applies a recent neural net architecture, BagNets, to evaluate the effect of different sizes of receptive fields in order to predict age and sex from MRI images. Results shows an accuracy of 4-5 years and 81-84% on sex classification. The interpretability of extracted predictive field are said to be unclear at this stage.
The paper seems to be results from an explorative work, however, without a comparative study, it is hard to judge on what could be improvements or not.
Table 1: the method describes receptive fields of size 9-17-33, is 177 also included?
Experiments: "with x,y, and z subject each", missing numbers from draft?
Interpretability on age: Aging is known to produce normal atrophy of the cortex, this is perhaps what is observed in Fig 1? On Sex classification, it may look like ventricule sizes are caught, is this motivated by existing literature?

---

### Official Review · AnonReviewer1 · 2019-05-01
**Direct application of a novel method with interesting results**

**Rating:** 3
**Confidence:** 2

**Review:**

This paper is a direct application of the recently published at ICLR 2019 paper "Approximating CNNs with Bag-of-local-Features models works surprisingly well on ImageNet", where the classification is not performed using convolutional layers, but instead using bags-of-local-features, that could be described as texture information of image patches, without spatial information. Good results using this method demonstrate that spatial information (and spatial relationship between several patches) might not be necessary for some tasks. Moreover, the method has a mode to highlight which patches are used for the predictions, which is good for interpretability.

The author apply it to sex (classification) and age (regression) prediction on 3D MR images, which required some trivial modifications to the original model (that was made for 2d images). An ablation study is performed based on the size of the receptive field (the patches size).

The results are interesting, since it works. This indicates that for this tasks, shape information might not be required. Another finding is that, on the contrary of the original paper, the results are more difficult to interpret, especially for the regression task (where there is a high variance in the predicted age between patches, superior to the life expectancy of a human).

Minor:
- It is not clear if the receptive field of size 177mm3 is the standard Resnet Baseline. I assume than yes, but this could be made explicit.
- For the age regression task, while the results are comparable wrt the receptive field size, performances still increase as we increase the receptive field size (which makes sense). However, we observe a sharp drop between 33 and 177 for the classification task (from 84.0% to 80.9%). Is there an interpretation about this ?

---

### Decision · Program_Chairs · 2019-05-06
**Acceptance Decision**

Accept